# Letea Virus: Comparative Genomics and Phylogenetic Analysis of a Novel Reassortant Orbivirus Discovered in Grass Snakes (*Natrix natrix*)

**DOI:** 10.3390/v12020243

**Published:** 2020-02-21

**Authors:** Alexandru Tomazatos, Rachel E. Marschang, Iulia Maranda, Heike Baum, Alexandra Bialonski, Marina Spînu, Renke Lühken, Jonas Schmidt-Chanasit, Daniel Cadar

**Affiliations:** 1Bernhard Nocht Institute for Tropical Medicine, WHO Collaborating Centre for Arbovirus and Hemorrhagic Fever Reference and Research, 20359 Hamburg, Germany; alex_tomazatos@yahoo.com (A.T.); maranda.iulia@gmail.com (I.M.); baum@bnitm.de (H.B.); bialonski@bnitm.de (A.B.); renkeluhken@gmail.com (R.L.); jonassi@gmx.de (J.S.-C.); 2Cell Culture Lab, Microbiology Department, Laboklin GmbH & Co. KG, 97688 Bad Kissingen, Germany; rachel.marschang@googlemail.com; 3Department of Clinical Sciences-Infectious Diseases, University of Agricultural Sciences and Veterinary Medicine, 400372 Cluj-Napoca, Romania; marina.spinu@gmail.com; 4Faculty of Mathematics, Informatics and Natural Sciences, Universität Hamburg, 20148 Hamburg, Germany

**Keywords:** arbovirus, reptile, orbivirus, Danube Delta, metatranscriptomics, surveillance

## Abstract

The discovery and characterization of novel arthropod-borne viruses provide valuable information on their genetic diversity, ecology, evolution and potential to threaten animal or public health. Arbovirus surveillance is not conducted regularly in Romania, being particularly very scarce in the remote and diverse areas like the Danube Delta. Here we describe the detection and genetic characterization of a novel orbivirus (*Reoviridae*: *Orbivirus*) designated as Letea virus, which was found in grass snakes (*Natrix natrix*) during a metagenomic and metatranscriptomic survey conducted between 2014 and 2017. This virus is the first orbivirus discovered in reptiles. Phylogenetic analyses placed Letea virus as a highly divergent species in the *Culicoides*-/sand fly-borne orbivirus clade. Gene reassortment and intragenic recombination were detected in the majority of the nine Letea virus strains obtained, implying that these mechanisms play important roles in the evolution and diversification of the virus. However, the screening of arthropods, including *Culicoides* biting midges collected within the same surveillance program, tested negative for Letea virus infection and could not confirm the arthropod vector of the virus. The study provided complete genome sequences for nine Letea virus strains and new information about orbivirus diversity, host range, ecology and evolution. The phylogenetic associations warrant further screening of arthropods, as well as sustained surveillance efforts for elucidation of Letea virus natural cycle and possible implications for animal and human health.

## 1. Introduction

The *Reoviridae* family is a large and diverse group of nonenveloped, icosahedral viruses with genomes composed of 9–12 linear molecules of double-stranded RNA (dsRNA). Reoviruses are divided between the *Spinareovirinae* subfamily (species with turrets on the core particle) and *Sedoreovirinae* subfamily (species with smooth, nonturreted core particles). They infect numerous host species, from plants to crustaceans, insects, aquatic and terrestrial vertebrates [1]. Among the 16 *Reoviridae* genera, the *Orbivirus* genus (subfamily: *Sedoreovirinae*) is the largest, having 22 species recognized by the International Committee on Taxonomy of Viruses (ICTV) and a significant number of species proposals [2]. Orbiviruses are vector-borne pathogens, primarily transmitted by ticks and other hematophagous insects (mosquitoes, *Culicoides* biting midges and sand flies). Their wide host range includes wild and domestic ruminants, camelids, equids, humans, marsupials, bats, sloths and birds [1]. The most studied orbiviruses are the *Culicoides*-borne *Bluetongue virus* (BTV, type species), *African horse sickness virus* (AHSV) and *Epizootic hemorrhagic disease virus* (EHDV), all known as important pathogens of livestock and wildlife [3]. Some orbiviruses such as Tribeč virus, Kemerovo, Lebombo and Orungo viruses have been detected in human infections and are considered human pathogens [4].

Orbiviral genomes consist of 10 linear segments of dsRNA designated by their decreasing molecular weight. They encode seven structural proteins (VP1–VP7) and three to four nonstructural proteins (NS1, NS2, NS3/NS3a and NS4) [1]. The high conservation degree of certain structural core proteins (e.g., polymerase, major core and subcore proteins) recommends them for comparative and phylogenetic analyses of different *Orbivirus* species [5,6]. In contrast, the proteins of the outer capsid are highly variable and their specificity to the host’s neutralizing antibody response can be used to distinguish between different serotypes of the same orbivirus species [7,8]. The phylogenetic clustering of *Orbivirus* members results in clades indicating their putative or potential arthropod vectors: *Culicoides*- or sand fly-borne (C/SBOV), mosquito-borne (MBOV) and tick-borne orbiviruses (TBOV) [9]. One exception to this classification is *St. Croix River virus* (SCRV), a distant member of the genus considered to be a “tick orbivirus” (TOV), having no known vector [5].

As one of Europe’s largest wetlands, the Danube Delta Biosphere Reserve (DDBR) located in the southeast of Romania, is a very biodiverse and heterogeneous complex of ecosystems [10]. The region is a major hub for bird migration along main African–Eurasian fly corridors, with ecoclimatic conditions suitable for abundant and diverse populations of arthropod vectors [11,12,13,14], which may allow pathogen import and maintenance [15,16,17,18].

During an arbovirus survey in DDBR, we identified a novel orbivirus in grass snakes (*Natrix natrix* Linnaeus 1758), tentatively named *Letea virus* (LEAV) after the eponymous village from the study area. The aims of this study were to characterize the genome of LEAV and its evolutionary relationship with other members of the *Orbivirus* genus. This is the first report of reptiles as orbivirus hosts. The present study expands our knowledge of orbivirus host range, ecology and the complete genomic data may help understand the evolutionary relationship among species of the *Orbivirus* genus.

## 2. Materials and Methods 

### 2.1. Snake Capture and Sampling

Clinical apparently healthy grass snakes (*n* = 190) and dice snakes (*Natrix tessellata* Laurenti 1768, *n* = 63) were captured by hand along transects in several areas of DDBR from 2014 to 2017, as part of an arbovirus survey (Figure 1, Appendix A**)**. A blood sample of ~1 mL was collected in a 2 mL sterile Eppendorf tube from the caudal vein of adults and subadults from both species (total *n* = 253). After clot formation and centrifugation for 2 min at 1000 rpm, the serum was carefully transferred to cryogenic vials using a 100 μL pipette with sterile filter tips. Samples were frozen at −28 °C in the field, shipped to the laboratory on dry ice and stored at −80 °C without interruption of the cold chain. All snakes were released immediately after blood collection back into their habitats. The DDBR Administration Authority issued research permits for all research activities (9/25.04.2014, 10692/ARBDD/25.04.2014; 7717/ARBDD/28.04.2016, 11/28.04.2016; 9/19.04.2017, 5627/ARBDD/13.04.2017).

### 2.2. Letea Virus Discovery and Genome Sequencing

The protocol used to perform metagenomic and metatranscriptomic on snake sera for virus discovery has been previously described [19]. Briefly, 100 μL sera serum samples used for deep-sequencing were filtered through a 0.45 μm filter (Millipore, Darmstadt, Germany) in order to remove larger debris and some bacteria. The filtrates were treated with a mixture of nucleases (Turbo DNase, Ambion, Carlsbad, CA, USA; Baseline-ZERO, Epicenter, Madison, WI, USA; Benzonase, Novagen, San Diego, CA, USA; RNAse One, Promega, Fitchburg, WI, USA) to digest unprotected nucleic acids including host DNA/RNA. Enriched viral particles were then subjected to RNA/DNA extraction by using MagMAX™ Viral RNA Isolation Kit (Life Technologies, Carlsbad, California, USA) according to the manufacturer’s instructions. After random RT-PCR amplification, the extracted viral RNA and DNA were subjected for library preparation by using a QIAseq FX DNA Library Kit (Qiagen, Hilden, Germany). Normalized samples were pooled and sequenced on a MiSeq or NextSeq550 platform. The generated raw reads were first qualitatively checked with Phred quality score <20 trimmed and filtered to remove polyclonal and low-quality reads (<55 bases long) using CLC workbench (Qiagen, Hilden, Germany). The remaining filtered raw reads were de novo assembled separately using Trinity v2.6.64239 and CLC workbench and compared with a nonredundant and viral proteome database (NCBI) using BLASTx with an E-value cutoff of 0.001. The virus-like contigs and singlets were further compared to all protein sequences in nonredundant protein databases with a default E-value cutoff of 0.001. The viral metagenomics and metatranscriptomics output have been visualized and analyzed in MEGAN [20].

### 2.3. Genetic Characterization and Phylogenetic Analysis 

Genome finishing, sequence assembly, and analysis were performed using Geneious v9.1.7. (Biomatters, Auckland, New Zealand) (Appendix A). Open reading frames (ORF) of the LEAV genome were detected with Geneious v9.1.7 and ORFfinder (https://www.ncbi.nlm.nih.gov/orffinder/). Putative functions of LEAV proteins were assigned by comparison to sequences in Genbank, using BLASTx. Pairwise distances for nucleotide and amino acid sequences were calculated in Geneious v9.1.7 using MAFFT.

Evolutionary relationship of LEAV with representative members of the *Orbivirus* genus were analyzed by inferring phylogenetic trees with amino acid and nucleotide ORF sequences of conserved genes encoding the polymerase (VP1), the subcore shell protein T2 (VP2/VP3) and the major core surface protein T13 (VP7) [5,6]. Nucleotide and amino acid sequences were aligned with MAFFT in Geneious v9.1.5. Phylogenetic analyses were performed with the best-fit nucleotide and amino acid substitution models selected by their lowest AIC (Akaike information criterion) or BIC (Bayesian information criterion) scores using jModelTest v2.1.10 [21,22] and ProtTest v3.4.2 [22,23], respectively. Amino acid phylogenetic trees were constructed using the maximum likelihood (ML) method in SeaView v4 [24] with LG+I+G+F for T2 and T13 (VP7) and for VP1 with the LG+I+G substitution models. The robustness of tree nodes was assessed by 500 bootstrap replicates. Nucleotide phylogenies were constructed by ML (500 bootstrap replicates) and by Bayesian Inference using the Markov Chain Monte Carlo (MCMC) method implemented in Beast v1.10.3 [25]. The output consisted of two combined MCMC chains, each of 10^7^ generations with sampling every 1000 steps and 20% burn-in. Figtree v1.4.3 (http://tree.bio.ed.ac.uk/software/figtree/) was used for visualization of tree output files.

Nucleotide sequences of LEAV genes obtained in the present study were submitted to GenBank and were assigned accession numbers MN873603–MN873692. The accession numbers of the other orbivirus sequences used for phylogenetic analysis are listed in Appendix A. 

### 2.4. Detection of Gene Reassortment and Intragenic Recombination

In order to screen LEAV for potential gene reassortment, we assembled complete genomes by segment concatenation and aligned them with MAFFT. Simplot v3.5.1 [26] was used to screen for potential reassortment between LEAV genomes (*n* = 9) using a 90% cutoff value for tree permutation across a given genomic segment. 

For detection of intragenic recombination, we inspected individual gene alignments in the Recombination Detection Program (RPD) package v4.95 and the tests therein (Bootscan, MaxChi, Chimaera, SiScan, PhylPro, 3seq and GENECONV) [27]. These tests were performed with default settings: a 200 bp window size and a Bonferroni correction of the *p*-value of 0.01. Recombination events were further considered upon detection of significant signals from at least three methods (Appendix A).

### 2.5. Screening of Potential LEAV Vectors

We retrospectively and concurrently analyzed arthropods collected within the same arbovirus surveillance program at the respective sites in DDBR (Figure 1), with the scope to identify a potential LEAV vector. In total, 18,093 *Culicoides* (16,829 unfed/gravid and 1264 engorged), 3973 engorged mosquitoes and 469 ticks were screened for detection of LEAV RNA (Appendix A). The unfed/gravid *Culicoides* (*n* = 16,829) and a part of the tick samples (*n* = 385) were screened using an RT-PCR assay. The engorged dipterans (1264 *Culicoides* and 3973 mosquitoes) and the rest of the ticks (*n* = 84) were subjected to metagenomic and metatranscriptomic analyses. 

The collection, processing and nucleic acid extraction from engorged mosquitoes and biting midges has been described in previous studies [14,28]. In the case of unfed/gravid *Culicoides* midges, insects were pooled as 1–118 specimens with the rest of the process being the same as in the above-referenced work. Ticks were collected from their hosts using fine tweezers and identified using morphological keys [29,30]. For homogenization, ticks were placed into a sterile 2 mL Eppendorf tube individually or as pooled nymphs, according to host, site and date of collection (6–9 specimens). Two 5 mm steel beads were added inside and the tube was then kept in liquid nitrogen for 1 min. The samples were loaded into a Tissue Lyser (Qiagen, Hilden, Germany) and the frozen ticks were pulverized at 50 Hz for 4 min. To each sample we added 0.6 mL of high-glucose (4.5 g/L) Dulbeco’s Modified Eagle’s Medium (DMEM) (Sigma–Aldrich, St. Louis, USA) with L-glutamine, 12.5% head-inactivated fetal bovine serum, 100 μg/mL streptomycin, 100 μg/mL penicillin and 100 μg/mL amphotericin B. The final mix was homogenized using the TissueLyser at 50 Hz for 2 min and clarified by centrifugation (10,000 rpm) for 2 min at 8 °C. Total RNA was extracted using the MagMax RNA/DNA Pathogen kit on a KingFisher Flex Magnetic Particle Processor (ThermoFisher Scientific, USA), according to the manufacturer’s instructions.

In order to detect the presence of LEAV in the reptile and arthropod samples, we designed a specific primer pair, 234F: AGGCAAAACAGTAGGATCAG and 234R: GGGCTAAGTGGATCTGAAAC, which amplifies a fragment of the outer capsid 2 protein (VP5). All PCR amplifications were performed in 10.8 μL consisting of 3 μL RNA, 4 μL reaction mix, 0.5 μL Mg_2_SO_4_ (0.25 μmol), 1 μL ddH_2_O, 1 μL of each primer (10 pmol) and 0.3 μL EnzymMix. The reactions comprised a first reverse transcription at 60 °C for 1 min, 50 °C for 45 min, 94 °C for 2 min, followed by 45 cycles of amplification at 94 °C for 15 sec, 55 °C for 30 sec and 68 °C for 30 sec. Final extension was at 68 °C for 7 min. RT-PCR was carried out using a Superscript III one-step RT-PCR kit (Invitrogen, Carlsbad CA, USA).

### 2.6. Isolation of LEAV

Sera samples from the LEAV RT-PCR positive snakes were used for attempted virus isolation on C6/36 (*Aedes albopictus*), Vero E6 (African green monkey kidney), BHK-21 (baby hamster kidney) and several reptile cell lines, including iguana heart cells (IgH-2, ATCC: CCL-108), Russell’s viper heart cells (VH-2, ATCC: CCL-140), *Terrapene* heart cells (TH-1, ATCC: CCL-50) and checked for viral cytopathic effect (CPE). Briefly, sera samples, undiluted and 1:10 diluted were inoculated onto the above-mentioned cell lines and were observed daily for cytopathic effects (CPE). All cultures were harvested 10 to 14 days later and subjected to the LEAV specific PCR test. This procedure was repeated until passage 5.

## 3. Results

### 3.1. Detection and Genomic Analysis of LEAV

Of the 190 *N. natrix* sera, 15 specimens of *N. natrix* (7.89%) were found positive for LEAV RNA. All samples collected from *N. tessellata* (*n* = 63) tested negative (Appendix A), as did all arthropod samples (ticks, mosquitoes and *Culicoides*) analyzed for the presence of LEAV RNA. Attempts to isolate the LEAV strains in several cell line cultures of different vertebrate and insect origins were not successful.

We obtained all 10 genomic segments of LEAV and assembled a total of nine complete genomes. BLASTx searches showed that the proteins encoded by LEAV genome match orbivirus homologs. Each segment is monocistronic with the encoded protein spanning most of the positive strand. One exception is segment-9, which additionally to VP6, contains a fourth nonstructural protein (NS4) on a smaller (+2) ORF. All nine LEAV strains have a genome length of 19,983 nucleotides and a GC content of 34.6–34.9%. Gene sizes range from 4010 bp (VP1) to 751 bp (NS3) and their coding asignments are homologous to BTV [1,31]. Descriptions of each LEAV gene with the closest relatives as retrieved by the BLASTx are found in Table 1. 

Sequencing of LEAV untranslated regions (UTRs) revealed that the segments share seven conserved nucleotides at both 3′ and 5′ termini. The first and last two nucleotides of all LEAV segments are inverted complements (Table 1) and identical to those found in most orbiviruses [1].

Comparison of main LEAV protein sequences to homologs of representative orbiviruses (Table 2) revealed identity values of 10–54% with *Culicoides*-/sand fly-borne orbiviruses (C/SBOV), 12–47% with tick and tick-borne orbiviruses (TBOV), and 11–46% with mosquito-borne orbiviruses (MBOV). The sequence identity between the polymerase (VP1) of LEAV and that of other orbiviruses was 33% (SCRV) to 54% (C/SBOV), above the 30% threshold proposed by [5] for viruses within a single genus of the *Reoviridae* family. Analysis of the VP3 protein indicated that it is the T2 protein forming the subcore shell, homologous to the VP3 of BTV and of other C/SBOV [31]. Similarly to VP1, the T2 is highly conserved and the level of identity relative to other orbiviruses ranged from 22% (SCRV) to 53% (C/SBOV). 

Segments 2 and 6 encode the outer capsid proteins VP2 and VP5 in LEAV (Table 1). VP2 is the most variable protein of C/SBOV, located in the first line of contact with host cells and a major determinant of virus serotype [31]. Significant levels of identity of the hypervariable VP2 were observed only between LEAV and insect-borne orbiviruses (IBOV, 10–15%), while within the same group VP5 comparison revealed values similar to T13 (Table 2). 

### 3.2. Phylogenetic Analysis 

The orbivirus VP1, T2 and T13 proteins are used in phylogenetic studies and for classification of *Reoviridae* members at both species and genus level [5,6]. LEAV was placed in the C/SBOV clade by all phylogenetic analyses, consistent with the levels of sequence identity revealed by comparisons with the other orbivirus proteins (Figure 2, Figure 3 and Appendix A).

The phylogenetic trees based on VP1 and T13 amino acid (Figure 2a,b) and nucleotide sequences (Appendix A) displayed a clustering typical for the *Orbivirus* genus, with main clades indicative of their arthropod vectors [9]. The three main branches are rooted by SCRV, an orbivirus isolated from tick cells which is considered a tick-associated orbivirus (TOV) [5].

The subcore shell protein T2 is encoded by segment 3, coresponding to the VP3 protein of the C/SBOV clade (Figure 2c). As in the VP1 and T13 phylogenies, LEAV is basal within this clade. The main difference is that the T2 tree splits between two clades instead of three, having a clear separation between the orbiviruses encoding T2 on segment-2 (MBOV, TBOV and SCRV (TOV)) and those encoding T2 on segment-3 (C/SBOV).

### 3.3. Detection of Gene Reassortment and Intragenic Recombination

Putative reassortment events involving LEAV segments were detected by Simplot (Bootscan) in eight of the nine LEAV genomes (Figure 3). Segment-7 (T13) was exchanged between NN23LRO17 and NN25LRO17, while the other instances indicated exchanges of segment-4 (VP4) and -9 (VP6) (Figure 3 and Appendix A). Significant signals of recombination among LEAV genes (≥3 methods) were found by RDP for segment-6 (VP5), segment-8 (NS2) and segment-9 (VP6) (Figure 3, Appendix A).

## 4. Discussion

Reptiles are known as hosts of numerous viruses. However, limited or fragmentary evidence is available regarding their role in arboviral cycles [32,33,34,35,36,37]. The present study investigated the possibility that natricine snakes harbor arboviruses by screening sera collected from sympatric populations of grass snakes (*N. natrix*) and dice snakes (*N. tessellata*) from the Danube Delta Biosphere Reserve in Romania. Thus, we described for the first time the discovery and genetic characterization of a novel orbivirus species (Letea virus, LEAV) infecting reptiles.

Inclusion and demarcation of species within the *Orbivirus* genus considers several criteria, such as sequence identity of segments encoding the polymerase (VP1) and major subcore shell protein T2, gene reassortment between close strains, high levels of serological cross-reactivity against conserved antigens like the T13 protein, conservation of UTR terminal nucleotides, range of hosts and vectors or the clinical signs associated with orbivirus infection [1]. We propose that LEAV should be included in the *Orbivirus* genus as a separate species, based on the comparative and phylogenetic analyses reported herein. In addition to a typical orbivirus genomic architecture of 10 linear segments of dsRNA, the UTRs of LEAV include conserved terminal sequences similar to other orbiviruses. The LEAV terminal nucleotides are not conserved hexanucleotides as in the case of BTV or AHSV, but heptanucleotides showing little variation among the 10 segments. Distal dinucleotides at both of the UTR ends are inverted complements (Table 1), as shown in other orbiviruses [1,3,6,9,38,39,40].

The NS4 is a nonstructural protein found in some orbiviruses and the last one described to date [41,42]. In LEAV, we found NS4 to be of similar size and position as in other C/SBOV [9,41,43], showing also the lowest sequence identity among all compared orbiviral proteins (Table 2).

The amino acid identity observed in the polymerase is above the 30% threshold defined by [5,44] for inclusion in the *Orbivirus* genus (Table 2). The protein sequence of LEAV T2 (VP3) showed identity levels significantly lower than the 91% cutoff indicated for this protein [5], confirming that LEAV is a distinct orbivirus species. Furthermore, the nine different LEAV strains belong to the same species, as their T2 amino acid sequences are >98% identical. Additional taxonomical markers of orbiviruses are the VP2 (outer capsid 1 in C/SBOV) and VP7 (T13) proteins, determining the serotype and serogroup, respectively [45]. The core surface protein T13 forms the outer layer of the viral core and is the primary antigenic constituent of virus serogroup (species) [31]. The low amino acid identity of LEAV T13 (Table 2) to other T13 proteins confirms that this virus belongs to a distinct serogroup. VP2 is encoded on segment-2 in LEAV and functionally equivalent to VP3 of MBOV and VP4 of TBOV/TOV [46]. Due to its neutralizing epitopes and role in cell attachment, the VP2 (OC1) protein is subjected to intense selective pressures by the host’s immune responses. Thus, it is one of the most variable orbiviral proteins [31]. Unsurprisingly, we found significant levels of identity for LEAV VP2 (10–15%) only in comparison with IBOV proteins. Moreover, the high amino acid identity (>98%) found between VP2 of the nine LEAV strains showed that all sequences belong to the same LEAV serotype. 

Previous studies noted that the overall GC content and the UTR proportion relative to the genome’s length reflect three groups similar to those illustrated by phylogenetic analyses. First, the GC content is highest in TBOV with 52–57.3% GC, followed by the C/SBOV with 39.9–45.9% GC and the MBOV with 36.7–41.6% GC [6,9,39,43]. The GC content of LEAV is 34.6–34.9%, therefore below these intervals. Second, the UTRs of C/SBOV span 3.5–4.1% of their total genome length, in TBOV the UTRs are between ~4.5–5% and in MBOV ~5–5.7% of the virus genome [6,47,48]. Again, LEAV falls outside these limits, having the proportion of UTRs at 6.64% of the genome’s length, which is higher than in other orbiviruses.

Apart from the high mutation rate owing to a polymerase lacking proofreading activity, reassortment of cognate genomic segments is an important driver of genetic diversity in viruses with segmented RNA genomes [49]. This process can generate novel phenotypes with fundamental implications for immune escape, host or vector range, virulence and pathogenicity [50,51,52,53]. The ability to reassort genomic segments is a primary criterion for inclusion in the *Reoviridae* family [31] and it may have contributed to the great evolutionary success of this family. Most natural cases of orbivirus reassortment have been described in BTV studies, mostly due to its antigenic diversity, wide geographic range, but also economic importance [54,55,56,57,58]. Additionally, reassortment has been described in EHDV [59,60,61,62], CORV [63], CGLV [64] and Banna orbivirus (BAOV) [65]. We found that reassortment between LEAV strains involved segment-4 (VP4), segment-7 (T13) and segment-9 (VP6) in eight of nine genomes analyzed. We speculate on another segment exchange where LEAV strain NN28SUL16 received its segment-3 (T2) from an unidentified LEAV parental strain, due to the striking sequence divergence of NN28SUL16 segment-3 within a highly similar (>98%) VP3 dataset (Figure 3 and Appendix A). When translated to protein sequence, the identity of this segment to the rest of the LEAV homologs was ≥98.9%, confirming the necessity of its structural conservation. 

Intragenic recombination between segments of LEAV was detected in the majority of LEAV strains (Figure 3, Appendix A). The effective contribution of each mechanism generating diversity in segmented viruses is far from being understood. Against a backdrop of rapid mutation, the formation of mosaic genes along with reassortment can have compounding effects on viral fitness. As in the case of sand fly-borne Changuinola (CGLV) serogroup [63], the strain biodiversity could be an important factor for the occurrence of RNA segment/fragment exchange in LEAV. This is indicated by detections of reassortment and intragenic recombination in eight, respectively seven, of nine LEAV strains. This is all the more clear for orbiviruses with great antigenic diversity like BTV [57,66,67] and AHSV [68], but also for orthoreoviruses [69,70,71] and rotaviruses [72].

Despite field efforts parallel to sera collection and the significant LEAV prevalence in grass snakes (7.89%), none of the screened arthropod pools or individuals tested positive for LEAV RNA. Also, recent analyses of mosquito and *Culicoides* host-feeding patterns in DDBR did not indicate ectothermic species as hosts of these insects (with very few exceptions provided by frogs fed upon by mosquitoes) [18]. Reoviruses known to infect reptiles belong to the turreted group of the family (subfamily *Spinareovirinae*, e.g., *Reptilian orthoreovirus*) and cause severe illness of digestive and respiratory organs [73,74,75,76]. The grass snake and dice snake occur sympatrically across the study area. In Dunărea Veche and Sulina we could observe some ecological features (bank structure, water body type, prey, microhabitat usage) very similar to those described for other European populations, indicating ecological partitioning in syntopic populations [77]. Although we encountered more *N. tessellata* in the aforementioned sites, LEAV was not detected in the sera of this species. Infected grass snakes showed no sign of disease. To our knowledge, this is the first report of orbivirus detection in reptiles.

All attempts of growing and isolating LEAV on insect or vertebrate cell lines showed no cytopathic effect or silent replication of the virus. Most known orbiviruses grow easily in vertebrate cells in vitro, while a few are known to be restricted to insect cells [78,79]. Although LEAV isolation could be further attempted on additional cell lines, we may speculate a shift in cell tropism underlying the inability of LEAV to infect certain cell types. A similar case is Parry’s Lagoon virus (PLV), a serotype of CORV. In contrast to the wide vertebrate host range of CORV [80], the antigenically related PLV showed a distinct cell tropism and replicated only in insect cells [81]. With the available data we may speculate that such shifts in cell tropism could be the result of successive changes through recombination, genetic drift and shift. *Reoviridae* is a very successful family of segmented dsRNA viruses, having wide host ranges across various econiches. The family’s repertoire of evolutionary strategies also includes deletion [82,83], gene duplication and concatemerisation [9,84,85]. These strategies were also observed previously in aquareoviruses [86], rotaviruses [87], phytoreoviruses [88] and even in a cross-family heterologous recombinant virus containing reovirus genomic components [89]. This suggests an increased potential for “species jumps” and adaptation to new vectors and/or hosts [81,86]. 

Earlier studies associated a conserved Arg-Gly-Asp (_167_RGD_171_) motif on the T13 protein of BTV with the attachment to *Culicoides* cells [90,91]. This conserved motif was also found by later studies in some species closely related to BTV [43,48,70], but we did not observed it in LEAV. As with other orbivirus species, this may reflect the higher divergence relative to BTV. Although the phylogenetic analyses indicate LEAV as a potentially *Culicoides*-borne orbivirus, it is interesting to note some general inconsistencies in vector associations of IBOV (C/SBOV and MBOV). For example, phylogenetic analyses place Orungo virus (ORUV), Lembombo virus (LEBV), Pata virus (PATAV) and Japanaut virus (JAPV) in the C/SBOV clade, even though they were discovered in mosquitoes [1,92,93]. For ORUV, follow-up studies on mosquito transmission were inconclusive [93]. Interestingly, Tracambé virus (TRV), a serotype of the sand fly-borne CGLV serogroup, was isolated from mosquitoes of the genus *Anopheles* (Reoviridae.org). Some serotypes of C/SBOV have been isolated from both mosquitoes and biting midges: Eubenangee virus (EUBV) [1,94], Palyam virus (PALV) [1,95], Warrego virus WARV [1], Wongorr virus (WGRV) [1], and Tibet orbivirus (TIBOV) [48]. The associations between some viruses and more than one vector family could be the result of “species jumps” permitted by fast evolution characteristic of RNA segmented viruses. Such occurrences in members of the *Orbivirus* genus would not necessarily run counter to the “coevolution” hypothesis [11,14,43,86], but possibly complement it.

In conclusion, a novel orbivirus (LEAV) was identified and characterized, expanding the known host range of orbiviruses and revealing its genetic relationship to the *Orbivirus* genus. Phylogenetic analysis indicates LEAV as a potentially *Culicoides*-borne orbivirus, although this was not confirmed by the screening of *Culicoides* midges and other arthropods from the DDBR. LEAV failed to replicate in vitro in several types of cells, which may warrant further attempts using additional cell types. The discovery and characterization of LEAV offers valuable information, expanding our knowledge about the evolution and host range of orbiviruses. The phylogenetic associations can justify further screening of arthropods and continued surveillance in order to describe the natural cycle of LEAV and its possible impact on vertebrate hosts.

## Figures and Tables

**Figure 1 viruses-12-00243-f001:**
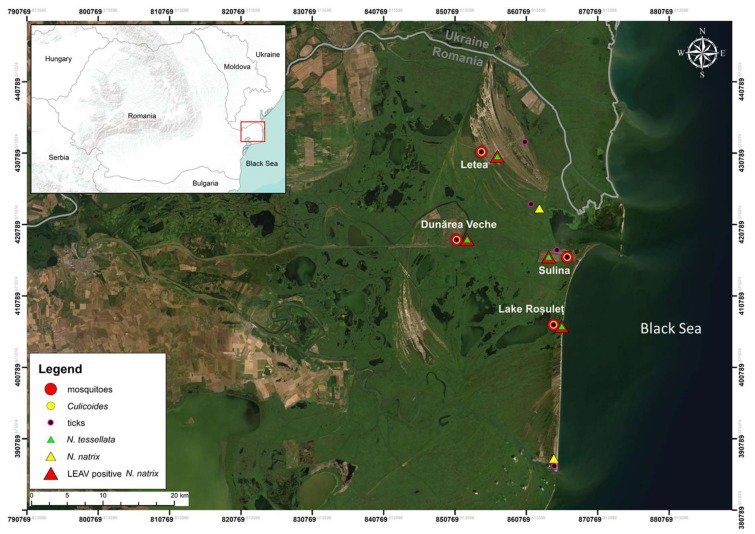
Satellite image of the study area (source: Esri) with sampling sites of grass and dice snakes and arthropods in Danube Delta Biosphere Reserve (Romania) during the study period 2014–2017. The various arthropod vectors were collected during 2014–2016 (mosquitoes), 2014–2017 (ticks) and 2017 (*Culicoides* midges).

**Figure 2 viruses-12-00243-f002:**
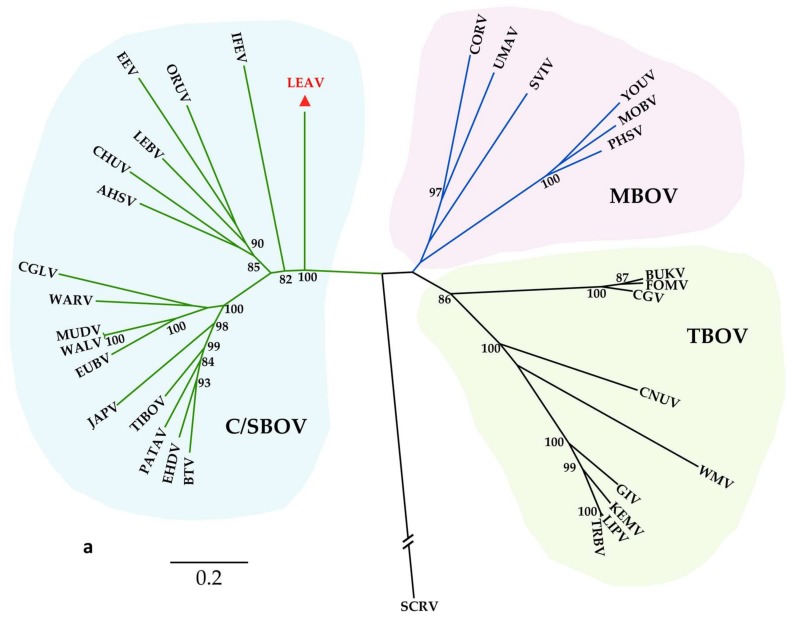
Maximum likelihood phylogeny of the VP1 (**a**), T13 (VP7) (**b**) and T2 (**c**) orbivirus proteins constructed using maximum likelihood inference and 100 bootstrap replicates. Bootstrap supports ≥80% are displayed at the nodes. Letea virus (LEAV) is indicated by the red triangle. C/SBOV stands for *Culicoides*-/sand fly-borne orbiviruses: African Horse Sickness virus (AHSV), Bluetongue virus (BTV), Changuinola virus (CGLV), Chuzan virus (CHUV), Epizootic hemorrhagic disease virus (EHDV), Equine encephalosis virus (EEV), Eubenangee virus (EUBV), Ife virus (IFEV), Japanaut virus (JAPV), Lebombo virus (LEBV), Mudjinabarry virus (MUDV), Orungo virus (ORUV), Pata virus (PATAV), Tibet orbivirus (TIBOV), Wallal virus (WALV), Warrego virus (WARV); MBOV stands for mosquito-borne orbiviruses: Corriparta virus (CORV), Mobuck virus (MOBV), Peruvian horse sickness virus (PHSV), Sathuvachari virus (SVIV), Umatilla virus (UMAV), Yunnan orbivirus (YOUV); TBOV stands for tick-borne orbiviruses: Bukakata virus (BUKV), Chenuda virus (CNUV), Chobar Gorge virus (CGV), Fomede virus (FOMV), Great Island virus (GIV), Kemerovo virus (KEMV), Lipovnik virus (LIPV), Tribeč virus (TRBV), Wad Medani virus (WMV); St. Croix River virus (SCRV).

**Figure 3 viruses-12-00243-f003:**
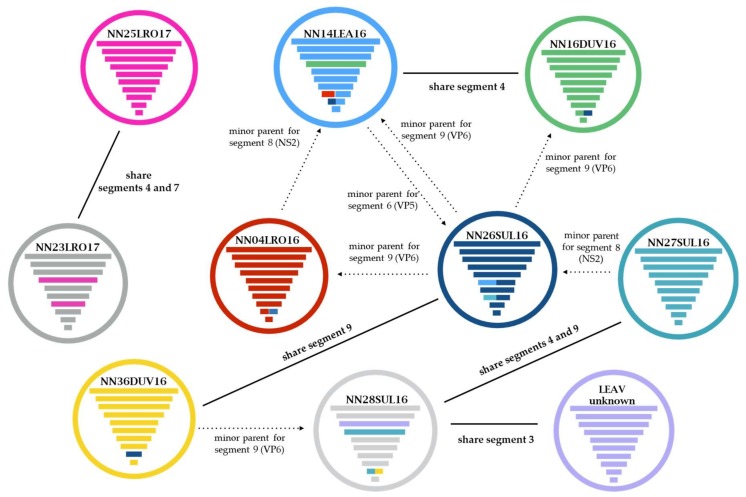
Graphical representation of gene reassortment and intragenic recombination between LEAV strains. Each colored circle represents a different LEAV strain. The 10 horizontal lines inside the circles represent the 10 genomic segments. Solid lines indicate reassortments of genes between different LEAV variants. Dashed arrows show the origins of gene fragments that have potentially been derived through recombination.

**Table 1 viruses-12-00243-t001:** Characteristics of the dsRNA genome segments of LEAV (NN04LRO16).

Segment	Protein Encoded	Segment Length (bp)	Segment ORF (with Stop Codon)	Predicted Protein (aa)	Predicted Protein Mass (kDa)	5’ UTR (bp)	5’ Conserved Terminus	3’ UTR (bp)	3’ Conserved Terminus	%GC	Top Blastp Results (ORF)% Pairwise Identity, Accession no.
**1**	RNA-dependent RNA polymerase (VP1)	**4010**	3855	**1284**	**147.84**	44	**5’-GUGAAAG**	111	**CAUUUAC-3’**	32.4%	VP1 AHSV-5 55.4%, AKP19848
**2**	similar to outer shell VP2 of BTV, neutralization epitope (OC1)	**3060**	2979	**992**	**115.13**	32	**5’-GUAAUUA**	49	**UUGUUAC-3’**	31.8%	VP2 BTV-5 25.6%, CAE51147
**3**	major subcore protein (T2/VP3)	**2851**	2727	**908**	**104.30**	68	**5’-GUAAAUG**	56	**GACUUAC-3’**	34.9%	VP3 LEBV 54.2%, YP_009507713
**4**	minor core and capping enzyme (CaP/VP4)	**2058**	1941	**646**	**75.50**	51	**5’-GUAAAAC**	66	**AAAGUAC-3’**	36.2%	VP4 PALV 50.5%, QCU80098
**5**	tubules (TuP/NS1)	**1960**	1851	**616**	**71.06**	29	**5’-GUAGAAG**	80	**GAUUUAC-3’**	37.0%	NS1 AHSV-8 32.1%, AKP19783
**6**	outer capsid protein (OC2/VP5)	**1684**	1605	**534**	**60.03**	41	**5’-GUAAAAA**	38	**GAAUUAC-3’**	36.0%	VP5 CGLV 48%, AGZ91957
**7**	major core surface protein (T13/VP7)	**1164**	1053	**350**	**39.30**	46	**5’-GUAUAAC**	65	**CACUUAC-3’**	37.7%	VP7 WALV 46.2%, AIT55708
**8**	nonstructural protein, viral inclusion bodies (ViP/NS2)	**1281**	1107	**368**	**41.04**	85	**5’-GUAAAUA**	89	**GACUUAC-3’**	36.8%	NS2 CGLV 33.5%, ACZ91977
**9**	minor core protein, helicase (Hel/VP6)	**1164**	936	**311**	**34.71**	61	**5’-GUAAUGA**	167	**AGCGUAC-3’**	33.6%	VP6 CGLV 32.1%, AGZ91984
	nonstructural protein (NS4)		246	**81**	**10.03**	-	**-**	-	**-**	40.2%	no results
**10**	nonstructural, virus release (NS3)	**751**	603	**200**	**21.62**	111	**5’-GUAAAAG**	37	**UCAUUAC-3’**	36.8%	NS3 IFEV 35.8%, QBL15286
**Total genome length**		**19,983**									

**Table 2 viruses-12-00243-t002:** Nucleotide (nt) and amino acid (aa) identities between LEAV and some representative orbiviruses: *Culicoides*-borne (AHSV, African Horse Sickness virus; WALV, Wallal virus), sandfly-borne (CGLV), mosquito-borne (PHSV, Peruvian horse sickness virus), tick-borne (CGV, Changuinola virus) and tick orbivirus (SCRV, St. Croix River virus).

Segment	Protein	AHSV		CGLV		WALV		PHSV		CGV		SCRV	
		nt	aa	nt	aa	nt	aa	nt	aa	nt	aa	nt	aa
1	VP1 (Pol)	58	**54**	58	**53**	60	**54**	56	**46**	50	**47**	42	**33**
2	VP2 (OC1)	30	**15**	31	**10**	33	**11**	31	**11**	19	**NSI**	21	**NSI**
3	VP3 (T2)	57	**52**	57	**53**	58	**53**	50	**36**	46	**37**	37	**22**
4	VP4 (CaP)	54	**50**	53	**49**	53	**48**	52	**44**	44	**40**	44	**35**
5	NS1 (TuP)	42	**26**	39	**24**	40	**22**	42	**21**	28	**16**	28	**15**
6	VP5 (OC2)	51	**41**	54	**46**	54	**45**	47	**34**	42	**31**	38	**27**
7	VP7 (T13)	52	**41**	52	**42**	53	**45**	41	**22**	37	**25**	32	**18**
8	NS2 (Vip)	42	**27**	45	**30**	45	**25**	34	**17**	34	**21**	25	**12**
9	VP6 (Hel)	36	**22**	38	**26**	37	**24**	43	**24**	33	**20**	24	**14**
	NS4	32	**NSI**	30	**11**	40	**NSI**	27	**NSI**	21	**NSI**	N/A	
10	NS3	35	**23**	37	**21**	36	**21**	32	**15**	36	**21**	29	**20**

NSI: no significant identity, N/A: not applicable.

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
