# Peer review of "Letea Virus: Comparative Genomics and Phylogenetic Analysis of a Novel Reassortant Orbivirus Discovered in Grass Snakes (Natrix natrix)"

_viruses, 2020, doi:10.3390/v12020243_

Round 1

Reviewer 1 Report

The manuscript of Tomazatos and colleagues gives a detailed and extensive examination of a novel Orbivirus, firstly described from reptiles. It is a well written manuscript with surprisingly extended examination of this novel virus. It gives good insight into the possible host range, ecology and evolution of Letea virus. It well uses the methods of interpretation to extend the diversity of Orbiviruses. In my opinion it is an important work of virology, and regarding the broad scale of host species of Reoviridae family, it is also an importan work in the context of emerging diseases. It works with high sample numbers, which were examined with state-of-the-art methods. I can only give minor recommendations before publication of this work.

Overall english is well written.

Minor recommendations:

It would be nice to see more details of in vitro isolation procedures (passage numbers, inoculation method, etc.)

Since metagenomic and metatranscriptomic methods were both used, it would be great to see detailed bioinformatic results (quality, read numbers, etc.) – maybe as a supplement.

It would be more informative, regarding the evolution of this virus to provide an additional Bayesian time-scale phylogeny. However I am familiar with the disadvantages of such analysis.

Author Response

Reviewer #1:

The manuscript of Tomazatos and colleagues gives a detailed and extensive examination of a novel Orbivirus, firstly described from reptiles. It is a well written manuscript with surprisingly extended examination of this novel virus. It gives good insight into the possible host range, ecology and evolution of Letea virus. It well uses the methods of interpretation to extend the diversity of Orbiviruses. In my opinion it is an important work of virology, and regarding the broad scale of host species of Reoviridae family, it is also an important work in the context of emerging diseases. It works with high sample numbers, which were examined with state-of-the-art methods. I can only give minor recommendations before publication of this work.

Response: We are very grateful to the reviewer for the constructive comments. We have made the changes and to accomodate according to your suggestions/requests. Please find below point-by-point responses to the issues raised.

It would be nice to see more details of in vitro isolation procedures (passage numbers, inoculation method, etc.)

Response: We have included additional informations regarding the isolation procedure and passage numbers.

Since metagenomic and metatranscriptomic methods were both used, it would be great to see detailed bioinformatic results (quality, read numbers, etc.) – maybe as a supplement.

Response:  We had included a supplementary table with the bioinformatic results, including the requested informations.

It would be more informative, regarding the evolution of this virus to provide an additional Bayesian time-scale phylogeny. However I am familiar with the disadvantages of such analysis.

Response:  

Response:  We agree that a time-scale resolution of the virus would give valuable information about the evolutionary aspects, e.g. the time to most recent common ancestor, but the amount of genetic data available is at the moment too limited to enable accurate phylogenetic clustering and resolution for the whole sandfly/culicoides-borne group in general, and for Letea virus in particular. One exception could be Blue tongue virus a very well characterized member of this group.

Reviewer 2 Report

The paper generally appears sound and is an interesting addition to the range of species that orbiviruses have been found in, it raises interesting questions about potential vector species with this more unusual host. I do however have several criticisms that need to be addressed before the paper can be accepted for publication 

I cannot access the sequences (the Genbank numbers listed do not return any results) and I would need to do so to verify any of the claims made in the paper 

I have some doubts as to your recombinant analysis, in part as it is not clear how you have obtained each "isolate" of virus - are these consensus sequences from an individual animal ie 9 individuals?  Is  it not possible that an individual animal could be infected with multiple variants?  You appear to have derived your sequences solely from short read assemblies with no long range (eg PCR and sangar sequencing) confirmation that these sequences are actually all on the one genome segment, this possibility at least requires discussion and some tempering of your conclusions 

Minor comment: Its not clear how many insects were in your insect pools or if any attempt was made to speciate your culicoides and mosquitoes (eg by morphological ID) before or after pooling

You must also have had viral reads other than the ones reported here? some comment on this (even if reported separately) would be appropriate. 

Author Response

The paper generally appears sound and is an interesting addition to the range of species that orbiviruses have been found in, it raises interesting questions about potential vector species with this more unusual host. I do however have several criticisms that need to be addressed before the paper can be accepted for publication

Response:  We are very grateful to the reviewer for the constructive comments. We have made the changes and to accomodate according to your suggestions/requests. Please find below point-by-point responses to the issues raised.

I cannot access the sequences (the Genbank numbers listed do not return any results) and I would need to do so to verify any of the claims made in the paper

Response:  We are sorry that the genomic data has not been yet released. This should happen in the next days as we informed the NCBI for immediately release.

I have some doubts as to your recombinant analysis, in part as it is not clear how you have obtained each "isolate" of virus - are these consensus sequences from an individual animal ie 9 individuals?  Is  it not possible that an individual animal could be infected with multiple variants?  You appear to have derived your sequences solely from short read assemblies with no long range (eg PCR and sangar sequencing) confirmation that these sequences are actually all on the one genome segment, this possibility at least requires discussion and some tempering of your conclusions

Response: The recombination results obtained have been confirmed by both mostly used recombination programs RDP and Simplot. We have obtained strong statistical support for the recombination events observed which is confirmatory. We do not named our strains as „isolate”, but strains. However, the genome finishing, sequence assembly, and analysis were performed using Geneious v9.1.5. This information has been added in the manuscript. Regarding the genomes obtained, we succesfully recovered 9 full genomes from 9 individuals, as we mentioned in the study. We absolutely agree that one individual could be infected in the same time with 2 ore more virus variants. However, we did not observed more than one virus variant in each individual nor SNPs. The genomes recovered using deep-sequencing had the coverage between 50x-260x which means a more than confortable coverage for complete genome recovery for short read seqeuncing. Thus, in this case the assembly and the contigs obtained for the genomes were consistent for fully genomic recovery. In addition, the metatranscriptomic data showed no other related Orbiviruses in the samples analyzed which exclude that some segments or parts of the virus genome belonged to other relatives present in the same individual.

Its not clear how many insects were in your insect pools or if any attempt was made to speciate your culicoides and mosquitoes (eg by morphological ID) before or after pooling

Response: Identification of midges and mosquitoes may have been unclear, as it was mentioned as “processing” in our manuscript and was divided between blood-fed  and unfed insects. Identification of the blood-fed insects was done per individual and referred to previous work: for blood-fed mosquitoes was done morphologically and is found in ref. [14]; for blood-fed Culicoides was done by barcoding and morphology and available in ref. [Tomazatos et al., 2020] .The unfed insects (only midges) were discriminated by their wing pattern, emphasising known vector groups (e.g. Pulicaris, Obsoletus groups). Unlike Pulicaris, the usual "Obsoletus" wing pattern is shared also by numerous non-Obsoletus group midges, so we decided to record them as “Other Culicoides”/“other Ceratopogonidae”, since we did not confirm presence of Obsoletus group members in the area. The main reason for this classification is the absence of key characters in many insects due to dry ice storage and not ethanol (intended for virus isolation).  Pooling for Letea virus screening was done only with unfed midges and the pools contained 1-118 insects (line 163) according to sampling date and site, subsequent steps and measurements being the same as for blood-fed midges [ref. Tomazatos et al., 2020]

You must also have had viral reads other than the ones reported here? some comment on this (even if reported separately) would be appropriate.

Response:  Indeed, beside the novel orbivirus species described in this study, there were other viral species from other viral families detected in those individuals. However, these findings are part of another study and there is no correlation with the results described here. That said, to mention the presence of other viral species is out of the aim of the present study and there would not bring any relevant information for better understanding the ecology, epidemiology and evolution of LEAV